# *Carcino-Evo-Devo*, A Theory of the Evolutionary Role of Hereditary Tumors

**DOI:** 10.3390/ijms24108611

**Published:** 2023-05-11

**Authors:** Andrei P. Kozlov

**Affiliations:** 1Vavilov Institute of General Genetics, Russian Academy of Sciences, 3 Gubkina Street, 117971 Moscow, Russia; contact@biomed.spb.ru; 2Peter the Great St. Petersburg Polytechnic University, 29 Polytekhnicheskaya Street, 195251 St. Petersburg, Russia

**Keywords:** *carcinoembryonic*, *evo-devo*, *carcino-evo-devo*, evolution, hereditary tumors, evolutionarily novel genes

## Abstract

A theory of the evolutionary role of hereditary tumors, or the *carcino-evo-devo* theory, is being developed. The main hypothesis of the theory, the hypothesis of evolution by tumor neofunctionalization, posits that hereditary tumors provided additional cell masses during the evolution of multicellular organisms for the expression of evolutionarily novel genes. The *carcino-evo-devo* theory has formulated several nontrivial predictions that have been confirmed in the laboratory of the author. It also suggests several nontrivial explanations of biological phenomena previously unexplained by the existing theories or incompletely understood. By considering three major types of biological development—individual, evolutionary, and neoplastic development—within one theoretical framework, the *carcino-evo-devo* theory has the potential to become a unifying biological theory.

## 1. Introduction

The theory of the evolutionary role of hereditary tumors, or the *carcino-evo-devo* theory, was developed in order to explain the sources of additional cell masses, which were necessary for the origins of new cell types, tissues, and organs during the progressive evolution of multicellular organisms. The content of the theory has been published in the theoretical papers of the author [1,2,3,4,5,6,7,8,9,10,11,12,13] and as a monograph in English [14], Russian [15], and Chinese [16]. Many experimental papers from the author’s lab were devoted to the confirmation of nontrivial predictions of the *carcino-evo-devo* theory [17,18,19,20,21,22,23,24,25,26,27,28,29,30,31,32,33,34,35,36,37,38,39].

The basic statements of the *carcino-evo-devo* theory are the following:Tumor processes participate in the evolution of development;Hereditary tumors provide evolving multicellular organisms with extra cell masses for the expression of evolutionarily novel genes and gene combinations, and thereby participate in the origins of new cell types, tissues, and organs;Populations of tumor-bearing organisms served as transitional forms in progressive evolution;Tumors may be considered search engines for new gene combinations in the space of biological possibilities.

Currently, the *carcino-evo-devo* theory consists of several parts describing different aspects of the theory. The main divisions of the theory include the following: (1) the analysis of the previous concepts; (2) the analysis of tumor features that might be used in evolution; (3) the main hypothesis (i.e., the hypothesis of evolution by tumor neofunctionalization); (4) nontrivial predictions and the results of their experimental confirmation; (5) nontrivial explanations of the *carcino-evo-devo* theory and the analysis of its relationship to other biological theories; (6) *carcino-evo-devo* diagrams; (7) the concept of tumors as search engines in the space of biological possibilities and their relationship to biological computational processes; and (8) tumors and the biocomputational formulation of the increase in complexity principle.

In this paper, the author will present the current state of the *carcino-evo-devo* theory by describing each of the theory’s principal divisions.

## 2. Preceding and Related Concepts

The term “*carcino-evo-devo*” was first introduced in [14], and it was used for the first time as a name for the theory of the evolutionary role of tumors in [9]. The new term was coined from two other terms: “*carcinoembryonic*” and “*evo-devo*”.

The term “*carcinoembryonic*” is used to designate embryonal proteins produced by tumor cells. The first examples of such proteins were alpha-fetoprotein [40,41] and carcinoembryonic antigen [42]. The concept of carcinoembryonic proteins was further developed in the concept of the convergence of embryonic and cancer signaling pathways [43,44,45,46]. These concepts point to molecular links between embryonal and tumor processes.

The abbreviation “*evo-devo*” corresponds to evolutionary developmental biology, which examines connections of normal individual development and evolution [47].

The *carcino-evo-devo* theory interconnects evolutionary, individual, and neoplastic development within one unified consideration. This theory studies the role of hereditary tumors in the evolution of development [9,14].

The previous concepts important for the *carcino-evo-devo* theory are the “embryonal rest” or “embryonal remnants” theory of cancer [48,49,50]; the morphological laws of the evolution of ontogenesis [51]; and the concept of neoplasia as a disease of differentiation [52].

Several previous ideas may be considered historical background for our theory, such as the idea of the role of “hopeful monsters” in evolution [53,54,55,56], and the idea of the positive role of viruses in evolution [57].

The *carcino-evo-devo* theory has been developing concurrently with Darwinian medicine [58], evolutionary epidemiology [59], and several branches of evolutionary oncology (e.g., the somatic evolution of tumor cells and selection in tumor cell populations [60,61,62], cancer selection [63], and the ecological hypothesis [64]). Comparative oncology has been in place since the 19th century (reviewed in [14]).

So, the *carcino-evo-devo* theory is deeply rooted in experimental oncology, developmental and evolutionary biology, and evolutionary medicine.

## 3. Tumor Features That Suggest the Role of Tumors in the Evolution of Organisms

There are features of tumors that suggest their evolutionary role. Many of them have been discussed in the author’s book [14]. Here, the author will briefly discuss this topic with the addition of new data and references, which were not cited in the book or appeared after its publication.

### 3.1. Hereditary Tumors, like Any Hereditary Trait, Could Be Used in Evolution

Many tumors are hereditary. The high incidence of certain tumors in different strains of laboratory animals is a well-known phenomenon [14,65,66]. Cellular oncogenes and tumor suppressor genes are inherited as part of germline DNA and thus provide a genetic basis for tumor inheritance. The importance of germline mutations in cancer was recognized after the pioneering works of Alfred Knudson [67,68,69]. Twenty percent of cancer-associated mutations occur in the germline DNA, and ten percent are both somatic and germline mutations [70,71]. There is an inverse association between germline susceptibility risk and somatic genetic variation in human cancer [72,73]. Heritable epimutations associated with hereditary tumors (e.g., in Lynch syndrome) have also been described [74].

Tumor prevalence in wild and captive animals can reach a high percentage (e.g., >50% in Santa Catalina Island foxes) [75,76]. The highest cancer prevalence has been recorded in species or populations with low genetic diversity, which suggests that it may have a genetic basis (reviewed in [75]). In humans, increased cancer prevalence is also associated with reduced genetic diversity [77,78].

Hereditary cancer syndromes connected with inherited mutations in cancer predisposition genes, such as *MMR* or *BRCA* genes, account for a considerable proportion (up to 10%) of all cancers in humans [79,80,81,82]. The identification and clinical management of patients with such mutations became routine in modern oncology [83]. For the purpose of our discussion, it is important that hereditary cancer syndromes are much more frequent than noncancer hereditary syndromes, the classical genetic diseases (reviewed in [84]).

Any hereditary trait may be selected and may potentially be evolutionarily meaningful. The fact that many tumors are hereditary suggests that they might participate in the evolution of organisms. For example, germline mutations in *BRCA1* and *BRCA2* tumor suppressor genes increase the risk of breast and ovarian cancer in humans, and *Brca1* and *Brca2* genes participate in the embryonic development and differentiation of mammary glands in mice (reviewed in [10]).

When I talk about the evolutionary role of tumors, I mean hereditary tumors. More discussion on hereditary tumors may be found in [14].

### 3.2. The Widespread Occurrence of Tumors in Multicellular Organisms

Tumors are widespread throughout the phylogenetic tree [14]. New reviews on this topic, which appeared after the publication of the author’s book, support the broad distribution of tumors across the tree of life [85,86].

Cellular oncogenes and tumor suppressor genes are also widespread throughout the phylogenetic tree, and many of them are ancient (oncogenes were already present in sponges). As we have shown in our recent paper [35], cellular oncogenes and tumor suppressor genes are the oldest gene classes, except for housekeeping genes. This means that tumors could have been present during the evolution of multicellular organisms since the very early stages.

The paper of interest to our discussion describes naturally occurring tumors in the pre-bilaterian cnidarian *Hydra*, with properties similar to cancer in vertebrates [87].

The widespread occurrence of tumors throughout the phylogenetic tree suggests that they may be evolutionarily meaningful.

### 3.3. A Considerable Portion of Tumors Never Kill Their Hosts

Many (maybe most) tumors never kill their hosts. Benign tumors are widespread in nature. They may represent more than half of all tumors [14]. New publications support this view: up to 80% of mammalian tumors are benign [76,88]. For example, in humans, benign lesions in the nasopharynx accounted for 75% of cases, with malignancies accounting for 25% [89].

At earlier stages of progression, tumors do not kill their hosts. Moreover, the overall direction of tumors in the early stages is toward regression [90]. Tumors can be selected for new functions at the early stages of progression [91].

The list of tumors that do not kill their hosts and could be used in evolution includes fetal, neonatal, and infantile tumors; carcinomas in situ and pseudodiseases; tumors that spontaneously regress; and sustainable tumor masses or extra cell masses in dynamic equilibrium with the organism [14].

When I talk about the evolutionary role of hereditary tumors, I mean benign tumors or tumors at the early stages of progression, or the kinds of tumors mentioned above, but not malignant tumors at the final stages of progression, which kill their hosts.

### 3.4. Tumors Have Many Features That Could Be Used in Evolution

Tumors are excessive cell masses that are not functionally necessary for the organism. Many unusual genes, which are not expressed in normal tissues, are expressed in tumors. There are also many unusual gene combinations expressed in tumors. Tumor cells can differentiate with the loss of malignancy, and tumors have morphogenetic potential.

All these features are much in demand in evolving multicellular organisms. However, they are not widely appreciated or discussed because the scientific community is more interested in tumors that kill the organism (i.e., malignant tumors, and the medical aspects of malignancy). On the contrary, in this paper, we are interested in tumors that do not kill their hosts, are inherited, and could be used in evolution.

The data exist that tumors were indeed used in evolution (e.g., the positive selection of tumor-associated genes in primate evolution (reviewed in [14])), and examples of organs that possibly originated from tumors [9,10,11,14,92]. The placenta [9,10,14], the mammary gland and prostate [9,10], and mammalian adipose [11] are a few examples of organs that might have developed from tumors. In the author’s book [14], further examples are covered, including the nitrogen-fixing root nodules of legumes, melanomatous cells and macromelanophores of *Xiphophorus* fish, hoods of Lionhead goldfish, and malignant papillomatosis and symbiovilli of voles.

New data suggest that tumors are connected with evolutionary adaptation [93] and the evolution rate [94].

### 3.5. “Nothing in Biology Makes Sense except in the Light of Evolution” [95]

Tumors are no exception to Dobzhansky’s maxima, and biologists have started thinking about the place that tumors occupy in evolution. Several monographs have been published over the last decade on tumors and evolution [14,63,96,97]. Until very recently, the role of tumors in evolution was considered completely negative. The “tumors and evolution” relationships were studied mainly in a comparative biological context, or in the context of the evolution of tumor cells in tumor cell populations. However, the hereditary nature of many tumors, the widespread occurrence of tumors, the fact that a considerable proportion of tumors never kill their hosts, and tumor features that could be used in the evolution of multicellular organisms are all features that suggest that hereditary tumors could play some positive role in the evolution of host organisms.

## 4. The Hypothesis of the Evolutionary Role of Hereditary Tumors (the Main Hypothesis)

The hypothesis of the evolutionary role of hereditary tumors was first formulated in [2], and it has since been developed in a series of articles [3,4,5,6,8].

In the book *Evolution by Tumor Neofunctionalization* (Chapter 10, Part 10.1), the hypothesis of the evolutionary role of hereditary tumors (or the “evolution by tumor neofunctionalization” hypothesis, or “main hypothesis”) was formulated as follows:

“Tumors are the source of extra cell masses, which may be used in the evolution of multicellular organisms for the expression of evolutionarily new genes, for the origin of new differentiated cell types with novel functions and for building new structures, which constitute evolutionary innovations and morphological novelties. Hereditary tumors may play an evolutionary role by providing conditions (space and resources) for the expression of genes newly evolving in the DNA of germ cells. As a result of the expression of novel genes, tumor cells may acquire new functions and differentiate in new directions, which may lead to the origin of new cell types, tissues and organs. The new cell type is inherited in progeny generations due to genetic and epigenetic mechanisms similar to those for pre-existing cell types. Tumors at the early stages of progression, benign tumors, pseudoneoplasms, and tumor-like processes, which provide evolving multicellular organisms with extra cell masses functionally unnecessary to the organism, are considered potentially evolutionarily meaningful. Malignant tumors at the late stages of progression, however, are not” [14]*.*

Figure 1 represents the graphical description of the hypothesis of evolution by tumor neofunctionalization.

Chapter 10 of the book [14] contains twelve more parts that discuss different aspects of the main hypothesis, including those represented in Figure 1. Evidently, the *carcino-evo-devo* theory was elaborated around the main hypothesis.

In Part 10.13 of the book, the author wrote:

“*Tumors could be regarded as unstable transitory search organs for innovations and expression of evolutionarily novel genes and new combinations of expressed genes, which are not possible in established organs…With the origin of new functions, atypical organs could be stabilized by functional feedback, accumulate a larger proportion of differentiated cells, and become new organs*” [14].

Thus, in the book [14], the evolutionary role of hereditary tumors was considered as providing conditions for the expression of evolutionarily new genes and gene combinations. By doing so, we reconciled the genetic theory of morphological evolution, which emphasizes the combinatorial changes in the expression of pre-existing genes [98,99,100,101,102,103,104], and the theory of evolution by gene duplication [105].

## 5. Nontrivial Predictions of the *Carcino-Evo-Devo* Theory

An important demand for any new theory in experimental science is the necessity to formulate nontrivial predictions that may be experimentally confirmed. The *carcino-evo-devo* theory has formulated such predictions.

The author’s previous article [38] was entirely devoted to nontrivial predictions of the *carcino-evo-devo* theory. The author examined several predictions of the theory and their confirmation in detail. That is why, here, we will discuss them only briefly.

### 5.1. The Number of Cellular Oncogenes should Correspond to the Number of Cell Types in the Organism [4,6]

This prediction was first formulated when only a few oncogenes had been described, and the anticipated number of oncogenes was estimated to be around twenty. At the same time, about 200 cell types had already been identified. So, the prediction of an order of magnitude larger number of oncogenes was quite bold.

Since the prediction was made, the number of oncogenes described has been continuously increasing. In our 2019 paper [35], we summarized the discovered numbers of oncogenes and cell types. In humans, the described oncogene number was between 245 (TAG database) and 380 [106], and the number of cell types was between 240 and 411, according to different authors (reviewed in [35]). A similar relationship was observed in other multicellular organisms [35].

So, as a result of the efforts of many laboratories, the prediction about the large number of oncogenes and its correspondence to the number of cell types was confirmed.

### 5.2. The Evolution of Oncogenes, Tumor Suppressors, and Differentiation Gene Classes Should Proceed Concurrently [14]

In [35], we showed that the gene age distribution curves of oncogenes, tumor suppressor genes, and differentiation genes overlap and form a cluster supported by hierarchical cluster analysis with perfect bootstrap reliability. The fact that the gene age distribution curves overlap means that oncogenes, tumor suppressor genes, and differentiation genes evolve concurrently, as predicted. Our results have been confirmed by other authors [71].

### 5.3. Evolutionarily New and Evolving Genes Should Be Specifically (or Predominantly) Expressed in Tumors (TSEEN Genes)

The laboratory of the author has devoted considerable effort to confirm this prediction. We have published many papers on this issue [12,17,18,19,20,22,23,24,25,26,27,28,29,30,31,32,33,34,35,36,37,38,39]. We have described individual genes, such as *PBOV1* [25,27,29] and *ELFN1-AS1* [23,26], and whole classes of tumor-specifically expressed, evolutionarily novel (*TSEEN*) genes, such as CT antigen genes [28] and ncRNA genes [32,35]. Many of our findings have been confirmed by other authors with references to our priorities [107,108,109,110,111,112,113,114]. In [38], the author suggested considering *TSEEN* genes as a new superclass of novel and evolving genes with tumor-specific or predominant expression, with several classes and families of *TSEEN* genes, which includes *TSEEN* genes of various phyla of organisms. A more detailed discussion of *TSEEN* genes may be found in a recent review by the author [38].

### 5.4. Human Orthologs of Fish TSEEN Genes Should Acquire Progressive Functions That Are Nonexistent in Fish

We studied this prediction using the *KRAS* transgenic inducible tumor model in fish [36]. The tumor after regression is a proxy for a new organ evolving from a tumor. The Gene Ontology method was used to investigate human orthologs of fish *TSEEN* genes that are expressed in tumors following regression. We found many orthologous human genes involved in the development of progressive traits in humans, such as the lung, mammary gland, placenta, ventricular septum, etc., that are nonexistent in fish [36]. The author suggested calling genes that originated from ancestral *TSEEN* genes and acquired progressive functions “*carcino-evo-devo*” genes. The discovery of *carcino-evo-devo* genes is a direct confirmation of the main hypothesis.

### 5.5. Selection of Tumors for New Functions in the Organism Is Possible

As a result of selection by Chinese breeders, a new organ, the hood, originated on the heads of certain varieties of goldfish. We have shown that such hoods are benign tumors [92]. Goldfish hoods have features of both normal organs (location, symmetrical shape, benign character, and appearance at certain stages of development) and tumors (the capability for unlimited growth and histological peculiarities). The author suggested calling such organs “tumor-like organs” [92].

Examples of the natural selection of tumors for new organismal functions are also known: symbiovilli in voles are the result of selection at the early stages of papillomatosis [91], and macromelanophores in swordtails are the result of sexual selection [115]. Many tumor-related genes in the primate lineage are positively selected (reviewed in [14]).

### 5.6. Evolutionarily Novel Organs Should Recapitulate Tumor Features in Their Development

The placenta, mammary gland, and prostate are evolutionarily novel to mammals. In [9,10,14], the author discusses many tumor features characteristic of these organs and classifies them as “tumor-like organs”. The term “tumor-like organ” has also been used for the placenta by other authors [116,117].

The ependymal region of the adult human spinal cord is different from that of rodents and primates and may be considered evolutionarily novel to humans. It shows ependymoma-like features morphologically and by gene expression profiling [118]. This example can be added to the list of mammalian tumor-like organs described in [10,11]. The discovery of D. Garcia-Ovejero and co-authors [118] means that, in humans, new cell types and tissues are being evolved, with the participation of tumor-like processes.

Unexpected results have been obtained with mammalian adipose. Mammalian adipose is a complex multidepot organ that consists of several adipose tissues and controls energy metabolism [119,120]. The adipose organ is evolutionarily novel to mammals because of its involvement in thermoregulation [121,122,123,124,125]. The author’s study of the tumor-like features of the mammalian adipose organ discovered many such features, including those belonging to the so-called “hallmarks of cancer” [11]. The most impressive similarities include the capability for unlimited expansion caused by cell hypertrophy and hyperplasia, chronic inflammation, and the infiltration of other organs, which are characteristic of obese adipose. This means that obesity is a tumor-like process, a conclusion that may be important for obesity prevention and treatment in the future. The author further studied whether the *carcino-evo-devo* genes described in [36] play any role in mammalian adipose, and discovered seven genes that participate in adipose development and form a gene network with mutual influences: *LEP*, *SPRY1*, *PPARG*, *ID2*, *CIDEA*, *NOTCH1*, and *ZAG* genes. Each of these genes, depending on context, also has tumor-promoting or tumor-suppressing functions [11,12,38]. Such an antagonistic bifunctionality may have fundamental importance for cancer and obesity problems.

### 5.7. Biological Examples of Cell Types, Tissues, and Organs Originated from Tumors

There are more biological examples of cell types, tissues, and organs that possibly originated from tumors: the nitrogen-fixing root nodules of legumes; the melanomatous cells and macromelanophores of *Xyphophorus* fishes; malignant papillomatosis and symbiovilli in the stomachs of voles (reviewed in [14]). The human brain recapitulates many features of tumors, especially during childhood and infancy [10,14].

Less characterized but interesting from the point of view of the main hypothesis are the set-aside cells of sea urchins and imaginal discs of insects; the ancestral neural crest and its derivatives; the anterior embryonic field that forms the right ventricle of the vertebrate heart; and morphological novelties and outgrowths in different species (e.g., the prepupal horn primordia of the *Onthophagus* species resembling papillomas (reviewed in [14])).

Some biological examples were suggested by the audience during the author’s presentations. This means that our concept captured the minds of biologists from different fields and did not contradict their professional experiences.

Malignant papillomatosis and symbiovilli in the stomachs of voles, described by N. Vorontsov [91], were suggested by the audience as examples during the author’s lecture at St. Petersburg University.

During the author’s presentations at Stazione Zoologia Anton Dorn, Naples, and the Zhirmunsky Institute of Marine Biology, Vladivostok, Italian and Russian zoologists independently suggested the following example and provided a reference: adults of sea urchins develop from asymmetrical tumor-like structures located at the left side (left coelomic pouch) of the larvae [126].

An outstanding example of the ependymal region of the adult human spinal cord [118] was advised by Dr. V. Korzh from Singapore University.

Biological examples are very important. In our case, it is difficult to find biological examples because tumor-bearing organisms that played a role in evolution are transient forms. The characterization of unstable elementary particles in physics demonstrates similar difficulties.

### 5.8. Conclusion about Nontrivial Predictions

The *carcino-evo-devo* theory has formulated several nontrivial predictions, which have been successfully confirmed. Thus, the *carcino-evo-devo* theory has predictive power, which is a fundamental requirement for a new theory.

## 6. Nontrivial Explanations of the *Carcino-Evo-Devo* Theory and Its Relationship to Other Biological Theories

The purpose of the theory is to explain unexplained (or not completely understood) phenomena, generate new knowledge, and create a scientific worldview. The creation of the new theory reflects the fact that previous theories do not completely resolve certain questions and thus do not satisfy the scientific community. The *carcino-evo-devo* theory provides a new explanation of the mechanisms of progressive evolution by taking into consideration hereditary tumors and tumor-like processes as active participants in complexity growth. For the first time, the *carcino-evo-devo* theory introduces the concept of relatively unstable transitional forms in evolution—tumor-bearing organisms—which explains the mechanisms of the origins of complex morphologies and multigene functions. The *carcino-evo-devo* theory adds the coevolution of neoplastic and normal development to the agenda of the *evo-devo* theory.

The *carcino-evo-devo* theory also explains several particular problems of the theory of progressive evolution, *evo-devo,* and other biological theories, from a new point of view. These explanations are discussed at length in [9,10,11,12,13,14] and are summarized in Table 1. As can be seen from Table 1, the *carcino-evo-devo* theory, in many cases, provides a new understanding of the existing problems, and it suggests experimentally testable mechanisms.

The *carcino-evo-devo* theory does not contradict Darwinism or the modern synthesis. In fact, the concept of selection is used throughout the author’s book [14] and in this paper (Section 5.5). However, the *carcino-evo-devo* theory is devoted to the origins of new cell types, tissues, and organs from hereditary tumors, which represent macroevolutionary events. Some authors believe that Darwinism and the modern synthesis, with their gradualism, do not explain macroevolution [54,55,56]. These authors suggest that different styles of genetic change, which include the rapid reorganization of the genome, are necessary [54,55,56]. Interestingly, the latter idea finds recent support in studies on the somatic evolution of cancer cells that demonstrate macroevolutionary catastrophic events, such as whole-genome doubling, chromosomal chromoplexy, and chromothripsis [141]. The genome architecture concept of H.H. Heng also considers cancer as emerging through macroevolutionary events, which is different from the somatic mutation theory of cancer [142,143].

The present author supposes that the *carcino-evo-devo* theory is complementary to Darwinism by providing a new mechanism of progressive evolution (i.e., evolution by tumor neofunctionalization). The *carcino-evo-devo* theory gravitates toward the concept of evolution determined by law, or nomogenesis, as described by Lev Berg [144].

Similarly, the *carcino-evo-devo* theory is complementary to the theory of evolution by gene duplication [105] in that it begins where the latter theory leaves off by failing to account for the origin of extra cell masses required for the expression of evolutionarily novel genes.

The *carcino-evo-devo* theory actively uses the ideas of the neutral theory of molecular evolution. Part 10.11 of the author’s book [14] is devoted to the nonadaptive origins of organismal complexity. The neutral theory of molecular evolution “regards protein and DNA polymorphisms as a transient phase of molecular evolution” [145]. In much the same way, the *carcino-evo-devo* theory regards hereditary tumors and tumor-bearing organisms as transient forms in progressive evolution. According to M. Kimura, “random genetic drift acting on selectively neutral mutations must have played some very important role in macroevolution, in particular in the evolution of gene duplicates” [146]. The *carcino-evo-devo* theory also deals with macroevolution. In [14], the author reviews the mechanisms of gene duplication and the neutral models of the evolution and maintenance of gene duplicates. Finally, random genetic drift acting on selectively neutral mutants may play a role in biological computational processes (see Section 8 below).

However, the *carcino-evo-devo* theory describes evolution at three structural levels of organization (molecular, cellular, and multicellular), as well as three main types of biological development (individual, evolutionary, and neoplastic), which sets it apart from the neutral theory, which is the theory of molecular evolution.

Many biological phenomena are explained by the theories that have been discussed above. The *carcino-evo-devo* theory makes synergistic links with the existing biological science branches by developing new explanations, and by providing a new mechanism of complexity growth (i.e., evolution by tumor neofunctionalization). The *carcino-evo-devo* theory fills in the spaces between and within the existing biological theories.

Representatives of numerous biological scientific areas largely agree with the *carcino-evo-devo* idea. As was already indicated, during the author’s presentations, the audience offered various biological examples.

Connections with many biological theories suggest the fundamental nature of the *carcino-evo-devo* theory. The *carcino-evo-devo* theory unites three main types of biological development—evolutionary, ontogenetic, and neoplastic development—in one consideration, and that is why it has the potential to become a unifying biological theory.

## 7. *Carcino-Evo-Devo* Diagrams

C*arcino-evo-devo* diagrams were introduced in [9], and more diagrams were added in [10]. At the moment, four diagrams have been published describing different aspects of the *carcino-evo-devo* theory [9,10].

The diagram in Figure 2 shows the interrelationship between three main types of biological development: individual, neoplastic, and evolutionary. From this diagram, it follows that normal ontogenies can participate in progressive evolution only through the intermediate transitionary stage, *carcino.* The *devo*→*evo* arrow is absent, which means that this particular process is prohibited because any deviation from normal development may be lethal (a developmental constraint). This is the fundamental prohibition—a prohibition of the saltatory origins of complex evolutionary innovations and morphological novelties, in which a multitude of genes are involved, from normal ontogenesis. Despite a strong resemblance to the central dogma of molecular biology, the first *carcino-evo-devo* diagram was built on an independent basis. However, the resemblance is striking and deserves a separate examination.

Figure 3 shows four successive steps in the evolution of ontogenesis. From this diagram, it follows that four different tumor processes are used at each of the successive steps of evolution. A more detailed description of the diagrams in Figure 2 and Figure 3 can be found in [9].

The diagrams published in [10] describe the role of tumor-like organs in the evolution of development and the origins of the placenta, mammary gland, and prostate in Eutherian ancestors [10].

The diagram of the central dogma of molecular biology and diagrams describing metabolic, immunological, and signaling pathways already exist and are widely accepted. *Carcino-evo-devo* diagrams may significantly add to this impressive diagrammatic description of fundamental biological connections. Mathematical category theory, in which objects are linked by arrows, may, in time, produce new generalizations in theoretical biology.

## 8. Tumors as Search Engines in the Space of Biological Possibilities and Biological Computation

The description of tumors as a “caricature” [147] or “parody” [148] of normal development, as well as the abnormal combination of normal cell features in tumors [52,149], have both sparked the idea that tumors can act as search engines. Tumors as search engines for novel molecular combinations are the focus of a specific section (Part 10.13) in [14].

In [13], the search engine nature of tumors was studied using the concept of biological computational processes [150,151,152,153,154] and Karl Popper’s concept of the possibility space [155].

As shown in [13], the search for new entities in the space of unrealized biological possibilities starts with the DNA computation of new genetic information. An evolutionarily novel gene may be considered a primary DNA computation. An evolutionarily novel gene that has not yet acquired a function may be considered Popper’s “reality in the making”.

The immunoglobulin locus of vertebrates demonstrates an example of DNA computation. Due to V(D)J recombination and subsequent mutations, a variety of immunoglobulins is created that corresponds to the variety of all possible antigens that vertebrates can meet [156,157].

The author suggested that, in much the same way, the evolving vertebrate genome can compute the whole space of possibilities for the evolution of vertebrate morphological structures and their adaptations to different environments [13].

However, DNA computation is not enough to generate the necessary diversity of immunoglobulins. As discussed in [9,13], additional cell masses are necessary (e.g., in amphibians). Additional cell masses can be provided by hereditary tumors [2,9,14].

Hereditary tumors also play the role of search engines in the possibility space of genetic information, which is the next step of biological computation [13]. Tumors search for new gene expression combinations. Many unusual genes and gene combinations, including *TSEEN* genes, are expressed in tumor cells due to their epigenetic peculiarities. Gene expression patterns that satisfy the compatibility rules described in [2] are frozen by functional feedbacks and natural selection [14].

## 9. The Increase in Complexity Principle

Earlier, the author discussed the fundamental nature of complexity’s increase at different levels of structural organization [2]. In the author’s book, this is further discussed in relation to the role of tumors in evolution [14]. The increase in complexity in progressive evolution has been discussed in terms of the expression of evolutionarily new genes and gene combinations in the extra cell masses of hereditary tumors [2,14]. In [13], the author discusses complexity growth from the point of view of computational biology.

The author suggests a biocomputational formulation of an old principle of the increase in complexity in living nature:

“The complexity increase in progressive evolution is realized through biological computation of the maximum number of compatible structural entities in evolving lineages of multicellular organisms. Biological computation of complexity increase involves DNA computation in the space of unrealized possibilities; stochastic gene expression and gene competition; compatibility search and incompatibility neutralization at different levels of organization; autonomous search engines and unfolding possibility spaces; unstable transitionary forms; and “freezing” of biologically meaningful constellations of entities, compatible within the ontogeny of multicellular organisms. The complexity of progressively evolving organisms tends to increase to a maximum and can be measured as the number of structural entities from the biological possibility space realized within the ontogeny of the multicellular organism” [13].

Briefly, in living nature, there is a tendency to realize the maximum number of compatible structural entities from the biological possibility space. This tendency is realized with the help of biological computational processes and autonomous search engines. The result is the increase in complexity of multicellular organisms in progressive evolution.

Thus, the concept of biological computational processes explains how complex multigenic functions and organs could originate, which otherwise would be difficult to imagine.

In [13], the author also formulated a new nontrivial prediction:

“After the origin of the genetic code, the evolution of DNA determined the evolution of living matter. That is why the existing informational network of DNA is not only for the storage of information about previous periods of evolutionary development. It also contains information about the future pathways of evolutionary development, which can be studied by bioinformatic approaches with the help of supercomputers. We can perform in silico computing of future functions of evolutionarily novel genes (in silico evolution), which are in the stage of their origin (Popper’s reality in the making) [149]. We plan to use machine learning algorithms to analyze the trajectory of TSEEN gene evolution between fish and humans. The obtained pattern of gene evolution will be used to predict the future functions of human TSEEN genes. Such computing will be similar to the calculation of unoccupied electron orbitals in physics” [13].

We can see that the *carcino-evo-devo* theory is actively developing, and its biocomputational part formulates new nontrivial explanations and predictions.

## 10. Conclusions

*Carcino-evo-devo* is a new biological theory that has predictive power and explains many unexplained phenomena. It suggests new ways of research and stimulates further thinking. As far as three major types of biological development—evolutionary, individual, and neoplastic development—are integrated in *carcino-evo-devo* consideration, this theory has the potential to become a unifying biological theory. This theory satisfies the criteria of beauty (*carcino-evo-devo* diagrams) and paradoxicality.

*Carcino-evo-devo* is also a new oncological theory that claims that hereditary tumors played a positive role in evolution, participating in the origins of new cell types, tissues, and organs. This is why oncological diseases may be considered a recapitulation of ancestral tumors, especially in evolutionarily novel tumor-like organs, such as the mammary gland and prostate. Using what we know about the co-evolution of tumors and normal development, we may understand how to stabilize tumors instead of trying to kill them by all possible means.

The *carcino-evo-devo* theory predicts the possibility of tumor stabilization by the enforcement of the functional feedbacks in the networks of *TSEEN* genes.

## Figures and Tables

**Figure 1 ijms-24-08611-f001:**
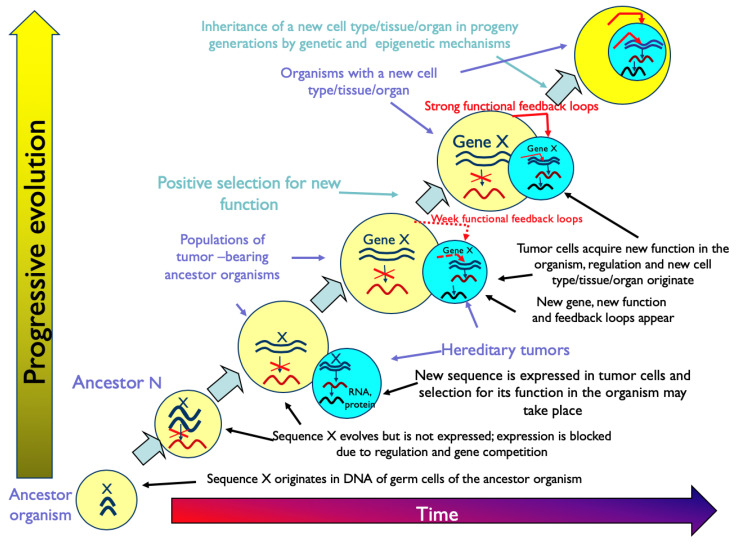
Graphical description of the hypothesis of evolution by tumor neofunctionalization. From [14], with permission. Copyright Elsevier (2014).

**Figure 2 ijms-24-08611-f002:**
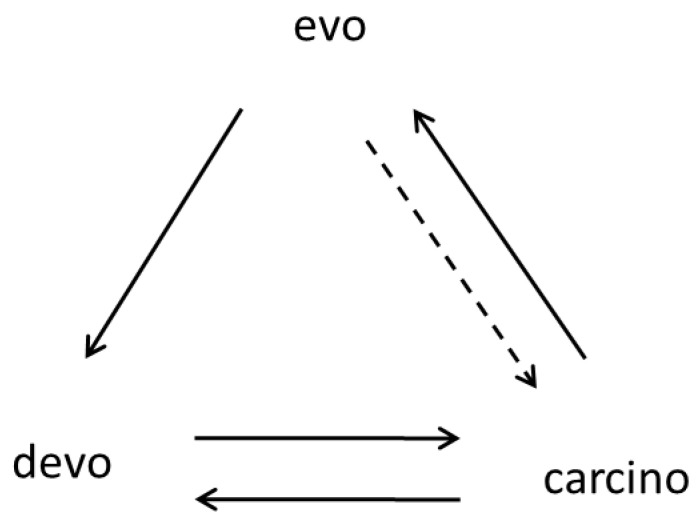
*Carcino-evo-devo* diagram showing the interrelationship between three main types of biological development: individual, neoplastic, and evolutionary (devo: normal ontogenies; carcino: ontogenies with neoplastic development; evo: progressive evolution of ontogenies). Arrows indicate participation in the corresponding process or essential connections. The dotted arrow (evo-->carcino) represents the evolutionary influence on neoplastic development (e.g., anticancer selection, [63]). From [9], with permission.

**Figure 3 ijms-24-08611-f003:**
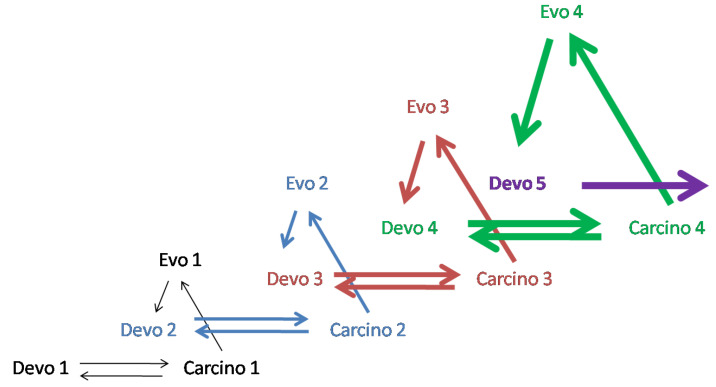
*Carcino-evo-devo* diagram showing four successive steps in the progressive evolution of ontogenesis (marked by different-coloured arrows). Four different tumor processes are used at each of the successive steps of evolution. From [9], with permission.

**Table 1 ijms-24-08611-t001:** Nontrivial explanations of the *carcino-evo-devo* theory.

What Is Explained	*Carcino*-*Evo-Devo* Theory Explanation	Other Existing Theories and Explanations
The nature of transitional forms in evolution, the origin of major morphological novelties, and complex evolutionary innovations in progressive evolution.	Populations of tumor-bearing organisms may represent transitional forms in progressive evolution. The morphological novelties and complex evolutionary innovations originate by tumor neofunctionalization [9,10,11,14].	The paleontological record is incomplete because periods of complexity growth were rare and of short duration [127]. Transitional forms are missing, both fossil and imaginary functional intermediates. The new theory is necessary [55,56]. Paleontologists do not expect to find “missing links”, as this is a misconception [128].
The possible way to overcome developmental constraints and developmental plasticity in evolution.	Developmental constraints are surmounted due to the increased plasticity of tumor cells and the transitory nature of tumor-bearing organisms and tumor-like organs [9,10,11,12,13,14].	Morphological novelties originated in progressive evolution despite the existence of developmental constraints [129]. The breaking of constraints may be associated with relaxed selection [130]. The mechanisms of transitions are not understood, and how developmental plasticity promotes innovations is unclear [131], but evolutionary explanations include developmental plasticity [132,133].
The common features of normal development and oncogenesis. The convergence of embryonic and neoplastic signaling pathways.	Common features of normal development and tumors, and convergence of embryonal and tumor signaling pathways, including carcinoembryonic antigens, follow from tumors’ participation in the evolution of development [9,14].	These similarities are usually explained by “aberrant” reactivation or the deregulation of embryonal pathways in tumors [134,135,136].
The source of extra cells for the origin of new cell types, and the origin of the neural crest determined cell types.	Our theory explains the origin of new cell types from tumor cells that are not functionally necessary to the organism [9,14]. The neural crest might originate from an ancestral embryonal tumor [14].	The existing theories of new cell-type origin (e.g., “sister-cell-type model” [137] or “serial sister cell type” [138]) do not explain how additional cells for the origin of the new cell types appeared.
The origin of new multigenic functions and organs.	Tumors are considered biocomputational search engines in the space of biological possibilities [13,14]. Tumors search for new gene expression combinations, thereby participating in the origins of new multigenic functions and organs.	The existing theory of gene regulatory networks [103] explains the development of existing organs and functions, but not the origin of new multigenic functions and organs.
The evolutionary role of tumors and cellular oncogenes, the phenomenon of CT antigens, and *TSEEN* genes.	The evolutionary role of tumors and cellular oncogenes might consist in providing evolving multicellular organisms with extra cell masses for the expression of evolutionarily novel genes and gene combinations [4,6,14,35]. CT antigen genes are evolutionarily novel, which is why they are expressed in the testes and in tumors [28]. *TSEEN* genes have been predicted by the main hypothesis [2,6,14] and described in the laboratory of the author [32,38].	To the best of the author’s knowledge, the *carcino-evo-devo* theory is the first theory that claims a positive evolutionary role for hereditary tumors. The author’s laboratory was the first to study the evolutionary novelty of CT antigen genes [28], later confirmed by other authors with reference to our priority [107]. *TSEEN* genes have also been confirmed by other authors with appropriate reference [113].
The evolutionary origin of the adaptive immune system.	The evolutionary role of tumors helps to explain the origin of the clonal expansion and clonal selection of lymphocytes, as well as of different immune cell types and organs. The ancestral tumor as a search engine could participate in the origin of the combinatorial joining of V, D, and J elements [9,13].	The existing theories of adaptive immunity’s evolutionary origins focus mainly on molecular events (e.g., the role of the RAG transposon and whole-genome duplications) [139].
The source of extra cell masses where evolutionarily novel genes with progressive functions are expressed.	Hereditary tumors might be the source of extra cell masses where evolutionarily novel genes and gene combinations with progressive functions are expressed [14,32,36].	The theory of evolution by gene duplication [105], as well as other theories of gene origin and genome evolution, do not address the cellular and multicellular aspects of the problem, focusing only on molecular events.
Peculiarities in structure and development of evolutionarily novel organs, and the higher incidence of tumors in such organs.	Tumor-like organs originated from ancestral hereditary tumors and recapitulate some of their features (the concept of tumor-like organs [10,11,12]).	J. Davis, who discovered the higher incidence of tumors in evolutionarily novel organs, such as the mammary gland and prostate, did not explain the phenomenon [140].

## Data Availability

Not applicable.

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
