# Peer review of "Carcino-Evo-Devo, A Theory of the Evolutionary Role of Hereditary Tumors"

_ijms, 2023, doi:10.3390/ijms24108611_

Round 1

Reviewer 1 Report

The concept of the role of hereditary tumors in evolution was put forward, which the author called the theory of carcino-evo-devo. This concept is based on the hypothesis of the acquisition of new functions (neofunctionalization) by hereditary tumors in the process of evolution, which, unlike cancerous tumors, are harmless to the body. The author postulates that such a transformation could provide additional cellular spaces for the expression of evolutionarily new genes emerging in the course of evolution of multicellular organisms.

The main statements of the carcino-evo-devo theory are as follows: (1) Tumor processes participated in the evolution of development. (2) Hereditary tumors provide evolving multicellular organisms with additional cell mass for the expression of evolutionarily new genes and combinations of genes, and thus participate in the emergence of new types of cells, tissues, and organs. (3) Populations of tumor-bearing organisms served as transitional forms in progressive evolution. (4) Tumors can be seen as search engines for new combinations of genes in the space of biological possibilities. Here the author uses the terminology  by well-known philosopher Karl Popper  and used by many contemporary authors for example Montévil, Maël. 2019. “Possibility Spaces and the Notion of Novelty: From Music to Biology.” Synthese 196 (11): 4555–81. https://doi.org/10.1007/s11229-017-1668-5. I am closer to the point of view expressed by the opponents of this concept, see for example for a recent discussion: Kauffman, Stuart, and Andrea Roli. 2021. "The World Is Not a Theorem" Entropy 23, no. 11: 1467. https://doi.org/10.3390/e23111467  : “that the morphological and behavioral evolution of the biosphere, besides the impossibility of being entailed is inherently not mathematizable in terms of sets. The World Is Not A Theorem…..”. However, the author here is involved in the current debate and his right to hold one or the other point of view.

The theory of carcino-evo-devo formulated several non-trivial predictions that were confirmed in the author's laboratory. In perspective, by considering the three main types of biological development—individual, evolutionary, and neoplastic—in one knot, the carcino-evo-devo theory could become a unifying biological theory. Carcino-evo-devo theory formulated several non-trivial predictions, (the author devoted a separate detailed table to the non-trivial conclusions of the theory),  which have been in this or that way confirmed. Thus, carcino-evo-devo theory has predictive power, which is a fundamental requirement to a new theory.

The very important, to my opinion, experimental achievement based on the concept is the discovery entire classes of tumor-specifically expressed evolutionarily novel (TSEEN) genes, specifically (or preferentially) expressed in tumors. The author proposed to consider TSEEN genes as a new superclass of new and developing genes, with several classes and families, in various types of organisms. It should be noted here that proving the novelty of a gene is an extremely difficult task. It is always unclear what level of evidence is necessary and sufficient to conclude that a gene is indeed of de novo origin. Second, it is difficult to determine the biological effects of putative de novo genes. The author and his colleagues performed this task with extreme care.

              I believe, that the main drawback of the manuscript is the author's somewhat exaggerated preoccupation with drawings in the form of diagrams. The manuscript contains 5 diagrams, which will be quite difficult for readers to understand, since the author does not follow the old recommendation for the presentation of figures: the figure should have such a caption that it can be understood without reference to the text. Signatures should be detailed so that it is immediately clear what it is. The number of figures should be reduced. At first glance it seems, for example, that figures 1 and 3 can be combined. But the author should think about other cuttings, otherwise the reader will get bogged down in wars with many arrows indicated evo1, evo2, evo 3 ….devo1, devo2, devo3 etc. .

But this shortcoming is easily eliminated. I believe that we are dealing with a very significant work that should be published.

Author Response

The number of diagrams is reduced to two diagrams; captions to Figs. 2 and 3 are extended in accordance with the recommendation of the Reviewer #1.

The author is grateful to the Reviewer #1 for his or her deep analysis of the manuscript 

Reviewer 2 Report

Although cancino-evo-devo biological theory is attractive biological theory, this reviewer wants to know the concrete example of this concept. Otherwise this theory is difficult to understand for the readers of the Int J med Sci.

** More detailed comments via email 4.8

• Do you consider the topic original or relevant in the field? Does it

address a specific gap in the field?

 This reviewer think the concept of A theory of the evolutionary role of

hereditary tumors, or carcino-evo-devo theory, is  interesting. However

the concept is diffecut to understan at a glance. Please describe the

difference of establised concept such as “Neutral theory of molecular

evolution”by Kimura M. (The neutral theory of molecular evolution and

the world view of the neutralists. Genome. 1989;31(1):24-31. doi: 10.

1139/g89-009. PMID: 2687096.) and Darwinism by Vendramin R, Litchfield K,

 Swanton C. Cancer evolution: Darwin and beyond. EMBO J. 2021 Sep 15;40(

18):e108389. doi: 10.15252/embj.2021108389. Epub 2021 Aug 30. PMID:

34459009; PMCID: PMC8441388.

• What does it add to the subject area compared with other published

material?

 If the above discussion is added, the manuscript woukd beneficial to

add to the subject area compared with other published material.

• What specific improvements should the authors consider regarding the

methodology? What further controls should be considered?

It would be helpful to understand the concept of the “the evolutionary

role of hereditary tumors, or carcino-evo-devo theory” in more graphical

image is described in Figures.

The author proposed to consider TSEEN genes as a new superclass of new

and developing genes, with several classes and families, in various

types of organisms. It should be noted here that proving the novelty of

a gene is difficult task.  The authors need to show the example of the

concept of TSEEN genes. It is always unclear what level of evidence is

necessary and sufficient to conclude that a gene is indeed of de novo

origin. Second, it is difficult to determine the biological effects of

putative de novo genes.

Abstract

A theory of the evolutionary role of hereditary tumors, or carcino-evo-

devo theory, is being developed. The main hypothesis of the theory, the

hypothesis of evolution by tumor neofunctionalization, posits that

hereditary tumors provided additional cell masses during evolution of

multicellular organisms for the expression of evolutionarily novel genes.

 The carcino-evo-devo theory formulated several non-trivial predictions

that have been confirmed in the laboratory of the author. It also

suggested several non-trivial explanations of the biological phenomena

previously unexplained by the existing theories, or incompletely

understood phenomena. By considering three major types of biological

development - individual, evolutionary and neoplastic development -

within one theoretical framework, the carcino-evo-devo theory has a

potential to become a unifying biological theory.

Review comment

Although cancino-evo-devo biological theory is attractive biological

theory, this reviewer wants to know the concrete example of this concept.

 Otherwise this theory is difficult to understand for the readers of the

Int J med Sci.

1.What is the significance of germline BRCA1/2 pathogenic variants in

hereditary breast and ovarian cancer (HBOC) patients and MLH1/MSH2/MSH6/

PMS2 in Lynch syndrome in individual, evolutionary and neoplastic

development? The example of HBOC and Lynch syndrome would be helpful to

understand the concept of  cancino-evo-devo biological theory for

readers of Int J med Sci.

2.Time course of “the evolutionary role of hereditary tumors, or carcino

-evo-devo theory” is required such as discussed in polygenic risk score

(PRS) in human evolution.

3.The author needs to discuss the “Neutral theory of molecular evolution

” by Kimura M. (The neutral theory of molecular evolution and the world

view of the neutralists. Genome. 1989;31(1):24-31. doi: 10.1139/g89-009.

PMID: 2687096.) and Darwinism by Vendramin R, Litchfield K, Swanton C.

Cancer evolution: Darwin and beyond. EMBO J. 2021 Sep 15;40(18):e108389.

doi: 10.15252/embj.2021108389. Epub 2021 Aug 30. PMID: 34459009; PMCID:

PMC8441388.

4.Is “The evolutionary role of hereditary tumors, or carcino-evo-devo

theory” different from the theory of “Neutral theory of molecular

evolution” and Darwinism? The authors need to describe the comparison of

the concept of this novel evolutional theory to understand the

difference of former major two evolutional concept as “Neutral theory of

molecular evolution” and Darwinism.

5.It would be helpful to understand the concept of the “the evolutionary

role of hereditary tumors, or carcino-evo-devo theory” in more graphical

image is described in Figures.

The author proposed to consider TSEEN genes as a new superclass of new

and developing genes, with several classes and families, in various

types of organisms. It should be noted here that proving the novelty of

a gene is difficult task.  The authors need to show the example of the

concept of TSEEN genes. It is always unclear what level of evidence is

necessary and sufficient to conclude that a gene is indeed of de novo

origin. Second, it is difficult to determine the biological effects of

putative de novo genes.

Author Response

First of all, the author wishes to thank the reviewer for helpful comments.

  1. English was edited by native English-speaking scientist and by using the online grammar checker.

  1. More examples of the concept are added to Part 2.4 of the manuscript (p.6, lines 9-13). Many biological examples of the concept may also be found in Part 4.7.

  1. The significance of BRCA1/2 genes in individual, evolutionary and neoplastic development is added (p.4, lines 21-24)

  1. The graphical image of the hypothesis of evolution by tumor neofunctionalization is presented on p.8.

  1. The comparison of the carcino-evo-devo concept with Darwinism and Neutral Theory is done: p.16, lines 3-6; p.17, lines 1-33; p.18, lines 1-2.

  1. TSEEN genes are addressed in Parts 4.3, 4.4, and 4.6 of the manuscript. In Part 4.4, the data is discussed that may be considered a direct confirmation of the main hypothesis of the carcino-evo-devo

In respect of difficulties in studies of the evolutionary novelty of genes, the best argument is the confirmation of our result by the other authors, which is mentioned in the manuscript.

The author published two reviews on TSEEN genes, one of them recently (Kozlov A.P. (2016), Expression of Evolutionarily Novel Genes in Tumors). Infect. Agents Cancer 11: 34; and Kozlov, A.P., The Theory of Carcino-Evo-Devo and its Non-Trivial Predictions. Genes 2022, 13, 2347 (https://doi.org/10.3390/genes13122347). Particularly, the de novo gene origin was discussed in the last review.

That is why, in the present manuscript, TSEEN genes are discussed only briefly. A sentence is added in Part 4.3 of the manuscript with a reference to the last review (p. 10, lines 13–14).

  1. The time course of events that the carcino-evo-devo theory describes is from the origin of multicellular organisms to the present day. This is covered in Part 2.2 of the manuscript.

The carcino-evo-devo theory has not addressed PRS as yet.

Round 2

Reviewer 2 Report

The authors have responded all the inquiries by this reviewer. This reviewer approved all the comments for the revised version of the manuscript entitled “Carcino-evo-devo, a theory of the evolutionary role of hereditary tumors” by Andrei P. Kozlov, is ready for publication consideration in editorial office of “IJMS”.

Author Response

The author wishes to thank Reviewer 2 for helpful comments that helped to improve the quality of the manuscript